# Risk of SARS-CoV-2 Reinfections in a Prospective Inception Cohort Study: Impact of COVID-19 Vaccination

**DOI:** 10.3390/jcm11123352

**Published:** 2022-06-10

**Authors:** José L. Casado, Johannes Haemmerle, Pilar Vizcarra, Gema Ramirez-Alonso, Andrea Salazar-Tosco, Beatriz Romero-Hernandez, Magdalena Blasco, Mario Rodriguez-Dominguez, Itria G. Mirabella, Alejandro Vallejo, Marina Fernandez-Escribano

**Affiliations:** 1Department of Infectious Diseases, IRYCIS (Instituto Ramon y Cajal para la Investigación Sanitaria), Hospital Universitario Ramón y Cajal, Ctra Colmenar Km 9, 28034 Madrid, Spain; pilar1vizcarra@gmail.com (P.V.); alejandro.vallejo@salud.madrid.org (A.V.); 2Department of Occupational Safety and Health, University Hospital Ramón y Cajal, 28034 Madrid, Spain; johannes.a.haemmerle@gmail.com (J.H.); gema.ramirez.alonso@gmail.com (G.R.-A.); andreacsalazart@gmail.com (A.S.-T.); magdalena.blasco@gmail.com (M.B.); itriagmirabella@gmail.com (I.G.M.); mfescribano@salud.madrid.org (M.F.-E.); 3Department of Microbiology, Centro de Investigación Biomédica en Red en Epidemiologia y Salud Pública, CIBERESP, Hospital Universitario Ramon y Cajal, 28034 Madrid, Spain; bromeroh@salud.madrid.org (B.R.-H.); mariojose.rodriguez@salud.madrid.org (M.R.-D.); 4Laboratory of Immunovirology, Department of Infectious Diseases, IRYCIS (Instituto Ramon y Cajal para la Investigación Sanitaria), Hospital Universitario Ramón y Cajal, 28034 Madrid, Spain

**Keywords:** SARS-CoV-2, COVID-19, immune response, reinfection, healthcare workers

## Abstract

The risk of reinfection could be related to the initial SARS-CoV-2 clinical presentation, but there are no data about the risk change after SARS-CoV-2 vaccination. We evaluated the rate of reinfection in an inception cohort study of 4943 health care workers (HCWs) according to symptoms and serologic results during March–May 2020. Incidence rates (IR) and IR ratios (IRR) before and after SARS-CoV-2 vaccination were determined by adjusting Poisson models. Overall, 1005 HCWs (20.3%) referred COVID-19 suggestive symptoms during the first surge of disease, and 33.5% and 55% presented a positive PCR or serology result, respectively. Meanwhile, 13% of asymptomatic HCWs had specific antibodies. During a follow up of 3422.2 person-years before vaccination, the rate of reinfection among seropositive individuals was 81% lower for those who were symptomatic compared with those who were asymptomatic (IRR of 0.19; 95% CI, 0.05–0.67; *p* = 0.003). During the 3100 person-years period after vaccination, an overall 74% decrease in the rate of infection was observed (IRR of 0.26; 95% CI, 0.21–0.32; *p* < 0.001), with a significant 83% and 70% decrease in seropositive and seronegative HCWs, respectively. In conclusion, the risk of SARS-CoV-2 reinfections is closely related to the clinical and serological presentation of COVID-19. COVID-19 vaccination further decreases the risk of reinfection more markedly among seropositive.

## 1. Introduction

Several studies have shown that the variability in clinical presentation and disease severity of COVID-19 is associated with individual immune responses to SARS-CoV-2 [1,2,3]. Although innate immune responses to SARS-CoV-2 appear to influence the extent of virus load and severity, the adaptive immune response plays a critical role in establishing an adequate immune evolution after infection [4,5].

In addition, most of the knowledge generated on immune responses and the duration of protection against SARS-CoV-2 is based on severe/hospitalized patients [6]. However, it has been estimated that 14–75% of infections are mild or asymptomatic [7,8,9]. In these patients, previous studies have demonstrated that the humoral immune response waned quickly after infection [1] and the extent of T-cell response seems to be lower in the case of asymptomatic or mild disease [1,4]. Considering data of seasonal endemic coronaviruses, we can expect that the waning of humoral immunity against SARS-CoV-2 and the probability of reinfection could be related [10]. Thus, monitoring of symptomatic and asymptomatic infection is necessary to assess the risk of reinfection. Nevertheless, to date, few studies have assessed the complex relation between symptoms, diagnostic tests, immune response, and the rate of reinfection.

Various studies have confirmed that healthcare workers are at increased risk of SARS-CoV-2 infection [11,12]. Thus, they could also have the highest risk of reinfection during the following surges of the disease, due to continuing work on the frontline against COVID-19. We designed an inception cohort study of healthcare workers (HCWs) to evaluate the relationship between the different clinical presentation and humoral response during the first wave of the disease, and to establish the incidence rates of reinfections before and after COVID-19 vaccination in seropositive individuals, compared to changes in seronegative HCWs.

## 2. Materials and Methods

This inception cohort study included 4943 individuals working during the first surge of COVID-19, starting in March 2020, at a tertiary university hospital in Madrid, Spain. A total of 6746 HCWs worked at that time at the hospital and had to be evaluated at the Department of Occupational Safety and Health in case of COVID-19 suggestive symptoms or direct contact with an indexed patient.

However, we finally selected 4943 HCWs (73%) that accepted to voluntarily participated in a seroprevalence survey starting before the end of April 2020, allowing us to evaluate the rate of occult infections and the seroprevalence of COVID-19, regardless of previous symptoms or infection [13]. We used these serological results to identify individuals diagnosed with COVID-19 not previously identified by polymerase chain reaction (PCR) tests. Although there were no differences in age, sex, or exposure, there was a slightly higher rate of PCR positivity among the 1803 employees not participating in the survey in comparison with our study population (11.3% vs. 7.2%), probably due to the greater interest of susceptible workers in participating in the serological screening.

For the included HCWs, exposure, epidemiologic, and demographic information was collected using a structured questionnaire, and the clinical, laboratory, and radiologic information was collected when performed. Frontline HCWs were defined as those who worked inwards and provided direct care to patients with confirmed or suspected COVID-19. We considered as suggestive symptoms of COVID-19 the acute onset of fever or chills, cough, shortness of breath or difficulty breathing, fatigue, muscle or body aches, headache, new loss of taste or smell, sore throat, or diarrhea. In these cases, nasopharyngeal swab PCR was performed and individualized advice about sick leave and quarantine was offered. The time since onset of symptoms, and the first positive and negative PCR results were collected.

For symptomatic HCWs, mild COVID-19 was defined as the absence of radiological infiltrates and lack of hypoxemia (oxygen saturation ≥94% on room air). Moderate disease was defined as the presence of symptoms attributable to COVID-19 with radiological infiltrates and oxygen saturation ≥94% on room air. Severe disease was defined as the presence of any of the following: oxygen saturation ≤93% at rest state; partial pressure of oxygen in arterial blood (PaO2)/fraction of inspired oxygen (FiO2) ≤300 mmHg (1 mmHg = 0.133 kPa) [14]. Due to the bias of HCWs being attended at home even in the case of more severe disease (or the bias of possible admission for better attention to colleagues), hospitalization was not considered as severe disease in the absence of other criteria.

We defined different study groups after the COVID-19 first wave according to the presence of suggestive symptoms (yes/no), PCR result (positive/negative), and specific serology results at April–May 2020 (positive/negative) (Figure 1):(a)Symptomatic HCWs with a confirmed disease (positive PCR OR/AND positive specific serology);(b)Symptomatic HCWs with positive PCR AND negative serology;(c)Symptomatic HCWs suggestive of COVID-19 but without laboratory confirmation (negative PCR AND negative specific serology);(d)Asymptomatic cases tested positive for viral RNA, due to close contact with a partner or an unknown patient, OR with a positive serology result in the early survey;(e)Asymptomatic patients who were not tested for viral RNA or were negative, and who had a negative serology result in the early survey.

Finally, all the included HCWs who continued working at the hospital were followed by the Department of Occupational Safety and Health until 15 November 2021 (before the detection of Omicron SARS-CoV-2 variant in our country), which encompassed the alpha (B.1.1.7) and delta (B.1.617.2) variant waves, and diagnoses of new infections or reinfections were collected. New infections were defined as a positive nasopharyngeal PCR result regardless of the presence of symptoms in those HCWs without previous positive PCR or serology, and they were collected to compare with the rate of reinfection before and after vaccination. Reinfection was defined as a positive PCR more than 90 days after the first viral RNA test, to exclude reactivation or recurrences [15], as established. Since the risk of new infections/reinfections is expected to differ according to vaccination, we evaluated the rate of infection in two different periods: from inclusion to SARS-CoV-2 vaccination (initiated at the end of January 2021 at our hospital), and from individual vaccination to 15 November 2021. To avoid possible bias (an even lower risk of a third episode, shorter follow up), new infections and reinfections during follow-up were excluded from further analysis.

### 2.1. Ethics Statement

According to the national guidelines on the obligatory occupational surveillance and privacy management, HCWs’ confidentiality was strictly safeguarded. Thus, data about HCWs were anonymized before analysis and established by alphanumeric code according to the protocols of the Department of Prevention of Occupational Safety and Health. The seroprevalence survey and follow-up was approved by our ethics committee (EC 249/20) with a waiver for written informed consent and was performed following the ethics standards noted in the 1964 Declaration of Helsinki and its later amendments.

### 2.2. Laboratory Procedures

Nasopharyngeal swabs were collected in a viral transport medium (Deltalab S.L., Barcelona, Spain) by trained healthcare staff and were processed in the same laboratory. During April and May 2020, serum samples were tested using the Vircell COVID-19 ELISA IgG and IgM/IgA tests (Vircell Spain S.L.U., Granada, Spain).

### 2.3. Statistical Analysis

Continuous variables are described as medians and interquartile ranges (IQRs). Categorical variables were described as frequencies and percentages. The Mann–Whitney U test, χ^2^ test, and Fisher’s exact test were used according to variable type as appropriate. We differentiated two periods, before and after vaccination, to calculate incidence rates per 100 person-years using Poisson regression analysis, and the incident rate ratio (IRR) and 95% CIs for COVID-19 infection in the different subgroups of HCWs. A 2-sided *p* < 0.05 was considered statistically significant. All analyses were performed using the SPSS statistical software version 20.0 (IBM Corp, Chicago, IL, USA).

## 3. Results

The overall distribution of COVID-19 diagnoses in the HCWs during the first wave is shown in Figure 1. Among 4943 HCWs, 1005 (20.3%) were evaluated at the Department of Occupational Safety due to COVID-19 suggestive symptoms, mainly fever, headache, cough, and anosmia (Table 1). Of these, 337 (33.5%; 95% confidence interval, CI, 31–37) had a positive PCR result, and 529 had specific antibodies against SARS-CoV-2. Therefore, early serologic determination in the follow-up improved the identification of COVID-19 patients, increasing the rate of confirmed diagnosis to 55%.

As shown in Table 1, most HCWs with COVID-19 experienced mild disease, 58 (10.9%) had a severe presentation, and no HCWs died because of COVID-19. Strikingly, 7.1% (95% CI, 4–10) of HCWs with positive PCR results had negative serology immediately after diagnosis. Although it was not statistically significant, these individuals had mild disease and were younger than the convalescent HCWs with positive serology, but the symptoms were similar. In addition, 452 (45%) symptomatic HCWs with suggestive symptoms had both negative PCR and serological results. In an in-depth analysis of this population, both fever and anosmia/ageusia were significantly less frequent, and the disease was more frequently mild. Additionally, 21 asymptomatic HCWs tested during the same period because of close contact with an index case had a positive PCR result (0.5% of asymptomatic). Finally, of 3917 asymptomatic HCWs participating in the serological survey without a COVID-19 diagnosis at the end of the first wave, 509 (13%; 95% CI, 12–14) had specific antibodies against SARS-CoV-2.

Thus, at the end of May 2020, 21.8% of the included HCWs had a confirmed diagnosis of SARS-CoV2 infection (11.1% symptomatic and 10.7% asymptomatic).

### 3.1. Infection/Reinfection before Vaccination

We were able to follow up the cohort during a median of 18.6 months (from April 2020 to November 2021, 6522 person-years) to ascertain the rate of new infections/reinfections before and after SARS-CoV-2 vaccination (Appendix A). Overall, 4597 (93%) HCWs continued working at the hospital, whereas the remaining 346 HCWs left the work after the first wave due to changes in labor conditions, without loss of follow-up attributed to the disease in any case. As shown in Appendix A, losses were homogeneously distributed among the different study groups. During a median follow-up of 268 days (3422.2 person-years) a total of 501 new infections/reinfections were observed between these HCWs (10.9%; incidence rate (IR) 14.6/100 person-years; 95% CI, 13.8–15.9, Figure 2). However, as shown in Figure 3, most of them were new infections in the group without previous symptoms, PCR, or positive serology (428/3161; 13.5%; IR 18.4/100 person-years, 95% CI, 17.1–19.8), and there were 18 reinfections among former seropositive HCWs (IR 2.48/100 person-years; 95% CI, 1.6–3.6). Thus, the rate of reinfections or new infections was 87% lower in seropositive than in seronegative HCWs (IR ratio, IRR, of 0.13; 95% CI, 0.08–0.21; *p* < 0.001) (Figure 4). Notably, among seropositive patients, there were no cases among symptomatic HCWs with previous positive PCR and serology (IR, 0/100 person-years; 95% CI, 0–1.29) whereas it was higher in previous asymptomatic seropositive cases (IR, 4.05/100 person-years; 95% CI, 2.6–6.2). Indeed, the rate of reinfection was lower in symptomatic than in asymptomatic seropositive HCWs (IRR of 0.19; 95% CI, 0.05–0.67; *p* = 0.003).

### 3.2. Infection/Reinfection after Vaccination

After vaccination with two doses of the BTN162b mRNA vaccine during January and February 2021, and after excluding those with a recent infection episode, 3821 HCWs were evaluated during a median time of 292 days (3100 person-years, Appendix A). Overall, 121 new infections/reinfections were observed during this period, a rate 74% lower than that observed before vaccination (IRR of 0.26; 95% CI, 0.21–0.32, *p* < 0.001 Figure 4). Importantly, the two doses of vaccine led to 83% (IRR of 0.17; 95% CI 0.05–0.57; *p* = 0.001) and 70% (IRR of 0.3; 95% CI, 0.24–0.36; *p* < 0.001) lower rates of reinfection for seropositive and seronegative HCWs, respectively, in comparison with the same groups before vaccination (Figure 4). Nevertheless, and despite the benefit of the vaccine, the differences in the rate of new infection/reinfection persisted between previously seropositive and seronegative HCWs (IRR of 0.07; 95% CI, 0.02–0.23; *p* < 0.001), and the higher IR was observed again in the previous seronegative, both symptomatic (IR 4.3/100 person-years; 95% CI, 2.6–7.03) and asymptomatic HCWs (IR 5.1/100 person-years; 95% CI, 4.33–6).

## 4. Discussion

Here, we showed the wide spectrum of SARS-CoV-2 infections a large, well-studied, prospective inception cohort of hospital employees when considering the combination of symptoms and diagnostic tests. We demonstrated that the different forms of presentation had immunological repercussions, in terms of the immune response and duration of the humoral response, as determined by the incidence of reinfections before and after vaccination.

Several studies have reported an increased risk of SARS-CoV-2 infection among healthcare workers [12]. The infection rate in these professionals ranged from 1.1% in China (0.74% if asymptomatic) [16], 3% in a group of asymptomatic workers at a UK teaching hospital [17], and 31.6% (half asymptomatic) in our milieu [18]. As of May 2020, we found that almost 22% of HCWs working in our hospital had confirmed COVID-19, and half of the cases also were asymptomatic. Although the incidence could be related to individual and occupational characteristics, a meta-analysis found that the prevalence of SARS-CoV-2 infection in symptomatic and asymptomatic patients was of 11% and 7%, respectively [19], similar to the rate found in our study (11.1 and 10.7%).

During the first wave of the disease, approximately one-fifth of the workers in our hospital consulted because of suggestive symptoms and they were tested. Of those, 35% of HCWs had a PCR-confirmed infection and 55% had a positive serology, representing 7% and 10.7% of the hospital workforce, respectively. However, nearly half of the participants with these symptoms showed negative results. This relatively low rate of positive PCR results has been described in other studies in the first wave. Kluytmans et al., found 6% of positive PCR results among 1353 workers with reported fever or respiratory symptoms [20]. Although the PCR false negative rate varies from 3% to 41%, according to the type of clinical specimen used, other situations including a delayed time to sampling and the procedure itself could be cause of an important number of false negative tests [21]. In line with this, anosmia was less frequently observed in a large percentage of symptomatic workers without an established final diagnosis (9%). Together with other symptoms, anosmia has been associated with a predictive value for diagnosing COVID-19, and it has been used to determine eligibility for community PCR testing when resources are limited [22,23]. Taken together, the absence of anosmia and ageusia and the negative PCR and serology suggest the possibility of an alternative etiology.

In this longitudinal cohort, we observed marked differences in the risk of infection/reinfection during the follow-up. As expected, the lowest risk was found in HCWs with previous symptoms and positive PCR and serology results, whereas the highest was found in asymptomatic seronegative individuals. Notably, the rate of reinfection in seropositive individuals was 87% lower than that observed in seronegative, and it was 5 times higher in those asymptomatic with positive serology in the survey than in symptomatic HCWs, suggesting a shorter duration of protection. In a longitudinal cohort, Lumley et al., reported that SARS-CoV-2 reinfections were rare, occurring also after mild or asymptomatic primary infection, and with an inverse correlation between positive antibodies and PCR-confirmed disease [24]. Thus, our data confirm the previous 80–90% of protection against reinfection observed in seropositive individuals in previous studies [24,25] and again suggest that the adaptive response associated with symptoms or more severe disease is important to avoid new episodes [4].

Strikingly, a small percentage of HCWs with symptoms and positive PCR results did not develop humoral response and had a lack of antibodies since the beginning of the disease. They represented 5–10% of symptomatic HCWs with PCR positivity and the ultimate reasons are not clear. We have recently shown that this fact could be related to a lower adaptive immune response in young patients with mild disease [4], two characteristics repeatedly described in this group. This fact could explain the differences observed in the duration of immune response, as only those with more severe disease are able to maintain a humoral immune response [3]. Indeed, we also observed a higher rate of reinfection, although not statistically significant, in this subgroup, and we have recently observed that this latter subgroup of symptomatic HCWs with positive PCR but lack of humoral response had a lower immune response to those with persistence of positive serology, and even a similar rate of response to that found in asymptomatic seronegative individuals [26]. Finally, the rate of infections in those HCWs with previous suggestive symptoms but negative COVID-19 tests was similar to the asymptomatic seronegative individuals, an indirect confirmation of absence of previous infection.

Nevertheless, in the period after vaccination we showed an overall 74% decrease in the rate of overall new infections/reinfections, and 83% and 70% of reduction among seropositive and seronegative HCWs, reducing the differences between groups. To highlight, seropositive individuals showed a nearly complete absence of reinfections, a fact that underlines the sum of protection achieved by the response to previous infection and vaccine. Indeed, even after vaccination the risk of new infections was 93% lower in those seropositive than in seronegative. Recently, we have demonstrated a rapid T-cell immune response to the vaccine in individuals with a previous disease or cross-reactivity, which is associated with adequate protection [26].

Our study has several limitations, in addition to the limitation of generalizing the infection incidence in relatively young and healthy people [27]. First, although sick leave had few personal financial consequences, testing was voluntary and was based on self-reported symptoms. This may have led to either overreporting or underreporting of symptoms. However, this fact is not expected to be substantial in this group of professionals with a high sense of responsibility. Similarly, reinfection testing was based on self-reporting of symptoms and those with past infection could have less suspicion of COVID-19, not asking for evaluation and therefore underestimating the risk of reinfection. Second, we did not include data about cycle thresholds in nasopharyngeal swabs of all the participants, and therefore a relationship between viral load, symptoms, and positivity of test was not performed. Finally, we did not adjust for initial values or changes in the titer of antibodies during follow-up which could modify the risk of reinfection. Also, we excluded cases of infection during follow-up to avoid the bias of changes in the risk of each group because of a recent infection.

In conclusion, the present study addressed the impact of the COVID-19 pandemic on HCWs in a large hospital during the successive waves of the disease, showing that the clinical presentation and the humoral response determine the risk of reinfection. Of note, vaccination increased this protection, more markedly in those individuals with past SARS-CoV-2 infection. Understanding the risks associated with reinfection in these different populations provides new opportunities for personalized risk stratification and reveals the correlates of protective immunity.

## Figures and Tables

**Figure 1 jcm-11-03352-f001:**
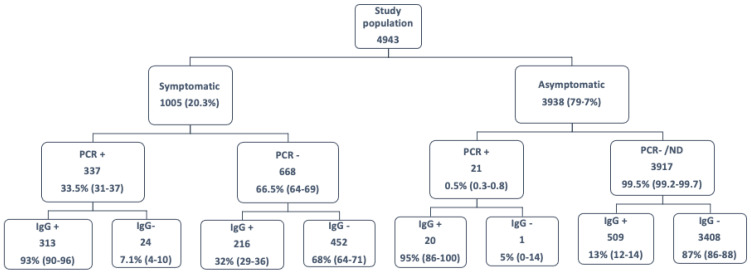
Flow diagram of 4943 HCWs included in the study after participation in the seroprevalence survey, classified according to the presence of symptoms, PCR result, and specific serology result (IgG+/IgG−). Between parentheses, 95% confidence interval of each rate.

**Figure 2 jcm-11-03352-f002:**
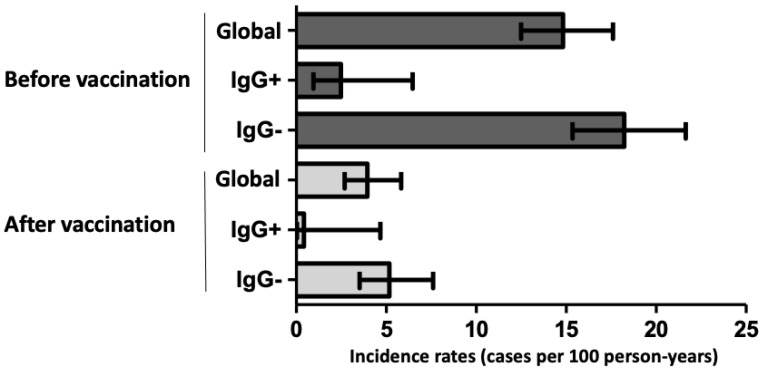
Incidence rates (95% confidence interval) of new infections/reinfections in the period before vaccination (dark grey bars) and after vaccination (grey bars), globally and according to positive (IgG+) or negative (IgG−) anti-N specific serology during the first wave of the disease.

**Figure 3 jcm-11-03352-f003:**
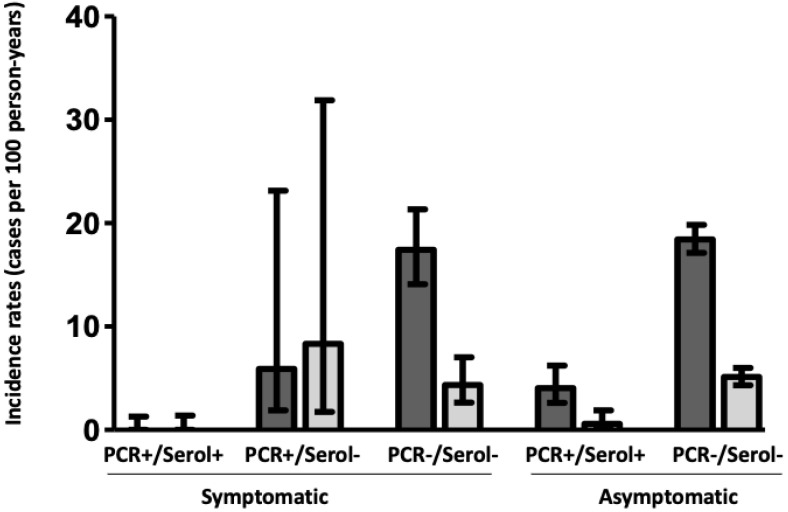
Incidence rates (95% confidence interval) of new infections/reinfections in function of the five different subgroups of the study (dark grey bars, before vaccination; grey bars, after vaccination). PCR, polymerase chain reaction; Serol, anti-N specific serology.

**Figure 4 jcm-11-03352-f004:**
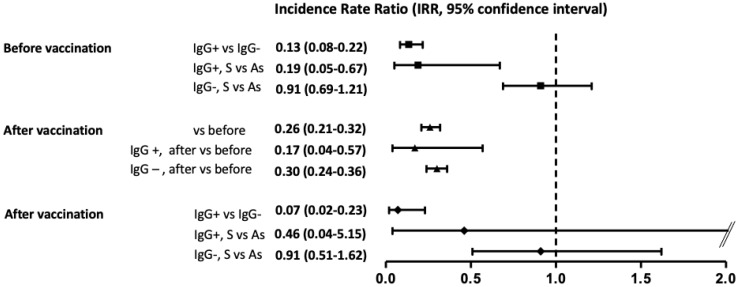
Incidence rate ratio (95% confidence interval) of new infections/reinfections according to period before or after vaccination, presence or no of specific antibodies, and classified as symptomatic or asymptomatic at the first surge of the disease. IgG, specific anti-N serology; S, symptomatic; As, asymptomatic.

**Table 1 jcm-11-03352-t001:** Clinical features of HCWs according to COVID-19 presentation in the inception cohort.

Variable		Symptomatic		Asymptomatic	
	PCR+/Serol+ (*n* = 529)	PCR+/Serol− (*n* = 24)	PCR−/Serol− (*n* = 452)	Serol+ (*n* = 529)	Serol− (*n* = 3408)
Female sex	398 (75)	12 (50)	371 (82)	296 (56)	1704 (50)
Age (years)	44 (32–60)	39 (22–52)	44 (33–61)	41 (23–59)	47 (24–60)
Comorbidities					
Hypertension	44 (8)	4 (13)	23 (5)	42 (8)	68 (2)
Diabetes	11 (2)	0	9 (2)	0	22 (0.6)
BMI (Kg/m^2^)	24.8 (22.1–26.2)	26.8 (23–27.3)	24 (23.5–26)	26.6 (24.1–27)	27.2 (24–29.1) *
COVID-19 frontline	365 (69)	13 (54)	271 (60)	335 (63)	1670 (49) *
Symptoms					
Fever	302 (57)	15 (63)	176 (39)		
Headache	249 (47)	9 (38)	199 (44)		
Anosmia	222 (42)	11 (46)	41 (9) *		
Cough	423 (80)	17 (71)	389 (86)		
Ageusia	102 (35)	9 (38)	41 (9) *		
Sore throat	233 (44)	12 (50)	267 (59)		
Severity ^a^					
Mild-moderate	471 (89)	24 (100)	434 (96) *		
Severe	58 (11)	-	18 (4)		

Data are presented as median, interquartile range, no. (%). * *p* value < 0.05 compared to PCR+/serol+ (left column), *p* values are calculated by *χ*^2^, Fisher’s test, or Mann–Whitney’s *U* test. Abbreviations: HCWs, healthcare workers; PCR, polymerase chain reaction; Serol, serology; BMI, body mass index; ^a^ Severity rating according to [14].

## Data Availability

Raw data will be available from the authors upon reasonable request.

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
