# Peer review of "Risk of SARS-CoV-2 Reinfections in a Prospective Inception Cohort Study: Impact of COVID-19 Vaccination"

_jcm, 2022, doi:10.3390/jcm11123352_

Round 1

Reviewer 1 Report

The manuscript describes a well-performed study on an important topic, i.e. the relation between clinical presentation and humoral response to COVID-19 infection, and the effect of clinical presentation, humoral response and COVID-19 vaccination on the incidence rate of (re)infection. The paper is well written, and the findings are relevant to the field. The observed seroprevalences of about 53% in symptomatic and 13% in asymptomatic frontline HCWs are in accordance with previous reports. The risk of (re)infection was - not unexpected - determined by COVID-19 vaccination. The most interesting and important finding is that the clinical presentation and humoral response determine the risk of (re)infection, both before and after COVID-19 vaccination. This finding underlines the sum of protection achieved by the response to previous infection and vaccine.  

Minor comments

1.     The description of the endpoint is not consistent. In the Methods section, Page 3, Line 100, the authors state that “… diagnoses of new infections or reinfections were collected …” during follow-up. However, both in the title and on several occasions in the body of the manuscript, the authors leave out the new infections. The authors are asked to consistently describe the endpoint used, i.e. “new infections and reinfections” or e.g. “(re)infections”.

2.     The conclusion stated in the abstract does not entirely reflect the objectives. In particular, a conclusion on the relationship between clinical presentation and humoral response is lacking. The same holds for the conclusions formulated at the end of the Discussion section.

3.     Abstract, Page 1, Line 20: It is not clear from this sentence that the 81% reduction that was observed when comparing symptomatic seropositives with asymptomatic seropositives. Please reword.

4.     Abstract, Page 1, Line 22-23: How do the authors explain an overall decrease of 74% where the decrease in both seropositives (83%) and seronegatives (78%) is higher?

5.     Methods, Page 2, Line 60: The description of the study population is not completely clear. It is stated that the “inception cohort included all individuals working during the first surge in March 2020 in the university hospital”. However, the authors continue to state that “6746 HCWS were evaluated for COVID-19 suggestive symptoms or direct contact with an index patient”. What percentage of the inception cohort was evaluated? If less than 100%, please report the percentage evaluated.

It is further stated that finally, 4943 HCWs who accepted to voluntarily participate were ‘selected’. Were all 6746 evaluated HCWs asked to participate? If no, based on what criteria were those 4943 HCWs selected? If yes, report the response rate (73%).

6.     Methods, Page 3, Line 107 / Discussion, Page 10, Lines 293-294: It is not entirely clear what the authors mean with excluding new infections and reinfections during follow-up in the subsequent analysis. Please explain.

7.     Ethics statement, Page 3, Line 112: Were data fully anonymized? The use of an alphanumeric code suggests that data may have been coded (pseudonymized).

8.     Ethics statement, Page 3, Line 114: It appears that ethics approval was only obtained for the seroprevalence study and not for the follow-up? Please confirm and explain.

9.     Ethics statement, Page 3, Line 115: It is suggested to include that the obtained waiver pertained to written informed consent. All HCWs in the analyses agreed to voluntarily participate, herewith providing oral consent.

10.   Results, Page 4, Line 140: The authors report that no HCW died because of COVID-19. It is unlikely that HCWs who died because of COVID-19 would have voluntarily participated in the seroprevalence study.

11.   Results, Page 4, Line 153: The observed 21.8% of HCWs with a confirmed diagnosis pertains to the HCW population studied, i.e. those with COVID-19 suggestive symptoms or direct contact with an indexed patient. The same holds for the observed distribution of symptomatic (11%) and asymptomatic (11%) HCWs among those with a confirmed diagnosis. This distribution might well not represent the distribution amongst all HCWs working in the university hospital.

12.   Results, Page 4, Line 153-154 / Discussion, page 9, Lines 235-243: By stating the overall rate of COVID-19 when including HCWs with COVID-19 suggestive symptoms, irrespective of the results of diagnostic tests, the authors suggest a high false-negative rate for PCR and/or serologic tests, instead of a low specificity of COVID-19 suggestive symptoms in the presence of other circulating respiratory viruses. Yet, the observed low frequency of COVID-19 specific symptoms (anosmia and ageusia) confirms the alternative aetiology of respiratory symptoms observed in HCWs with negative PCR and COVID-19 serology. It is suggested to leave out this hypothetical calculation from the Result section and to rephrase the the discussion of false-negative results in the Discussion section.  

13.   Discussion, Page 10, Lines 286-288: The authors mention self-reporting of symptoms and therewith the risk of underestimation of the risk of reinfection as a limitation of the study. This may indeed have biased the results. Was any serologic testing performed during the first follow-up period, i.e. until vaccination, to have an indication of this effect?

14.   Discussion: Serologic response was qualitatively measured. Quantitative assessment of humoral responses may have provided an explanation for the observed effect of clinical presentation on the risk of (re)infection (i.e. lower antibody titers in asymptomatic cases / higher (re)infection rates with lower antibody titers). The lack of quantitative data may be considered a limitation of the study and is suggested to be added as such to the Discussion section.

15.   Discussion, Page 10, Lines 295-300: It is questioned whether the first statement of the conclusions reflects the paper (“ the present study addresses the impact of the COVID-19 pandemic on different groups of HCWs”). It is suggested to phrase the conclusions to reflect the findings of the study. Although evident, adding the finding that vaccination (in addition to clinical presentation and humoral response) determines the risk of (re)infections might improve the completeness of the message.   

Reviewer 2 Report

Brief summary

The topic of this manuscript, the rate of SARS-CoV2 reinfection in accordance with clinical presentation and produced immunity is of very high interest since this will be the case in the upcoming waves of the pandemic. The authors presented their data in a well-structured manner.  Although they present data in a relatively young and healthy group of people (HCPs) they give an adequate risk reduction of reinfection in the general population before and after vaccination. Major limitation of the study is that there is no measurement of changes of adaptive immunity nor in the titer of antibodies (anti-spike or better neutralizing), neither in T-cell function.

The cited references are mostly recent publications and relevant. It includes several self-citations, but it seems necessary to advocate the results.

Finally, the conclusions are consistent with the evidence and arguments presented.

Specific comments

In the introduction

Line 29: rephrase to get clear which disease you are referring

Lines 31-33: There is no clear rationale for this statement, nor the references made it clear

In the Materials and Methods

No comments. Appropriate and well described

In the Results

line 142: “younger” is this significant? what is the p value?

Line 145: “ significantly less frequent anosmia/fever” This statement isn’t confirmed nor in this field neither in the Table

Lines 153-155: No need for this sentence

In Table 1

Lines 193-195: Even though the abbreviation section mention p values these aren’t presented in the table. p values must be presented in the table.

In Figures

No comments. Proper and accurate

In the Discussion

Lines 218-220: Since this paper doesn’t analyze clinical presentations this must be rephrased

Line 270: lower or similar? can’t be both

Line 292: “could produce” better rephrase
